# Unleashing the Full Potential of Oncolytic Adenoviruses against Cancer by Applying RNA Interference: The Force Awakens

**DOI:** 10.3390/cells7120228

**Published:** 2018-11-23

**Authors:** Tereza Brachtlova, Victor W. van Beusechem

**Affiliations:** Amsterdam UMC, Vrije Universiteit Amsterdam, Medical Oncology, Cancer Center Amsterdam, De Boelelaan 1117, 1007 MB Amsterdam, The Netherlands; t.brachtlova@vumc.nl

**Keywords:** RNA interference, small interfering RNA, microRNA, oncolytic virotherapy, conditionally replicating adenovirus (CRAd)

## Abstract

Oncolytic virus therapy of cancer is an actively pursued field of research. Viruses that were once considered as pathogens threatening the wellbeing of humans and animals alike are with every passing decade more prominently regarded as vehicles for genetic and oncolytic therapies. Oncolytic viruses kill cancer cells, sparing healthy tissues, and provoke an anticancer immune response. Among these viruses, recombinant adenoviruses are particularly attractive agents for oncolytic immunotherapy of cancer. Different approaches are currently examined to maximize their therapeutic effect. Here, knowledge of virus–host interactions may lead the way. In this regard, viral and host microRNAs are of particular interest. In addition, cellular factors inhibiting viral replication or dampening immune responses are being discovered. Therefore, applying RNA interference is an attractive approach to strengthen the anticancer efficacy of oncolytic viruses gaining attention in recent years. RNA interference can be used to fortify the virus’ cancer cell-killing and immune-stimulating properties and to suppress cellular pathways to cripple the tumor. In this review, we discuss different ways of how RNA interference may be utilized to increase the efficacy of oncolytic adenoviruses, to reveal their full potential.

## 1. Oncolytic Virotherapy

Despite many different types of treatments used to combat cancer, cancer is still one of the leading causes of death worldwide. While on one hand the specific traits cancer cells acquire, such as infinite replicative potential and evading apoptosis [1], provide angles for anticancer treatment, these traits on the other hand complicate treatment and, in many cases, contribute to treatment failure. Constant replication and genetic instability allow for adaptation to medication and emerged resistance. There is, therefore, a great need for new treatments that are also effective against therapy-resistant cancers. One promising approach is oncolytic virus therapy (OVT), using viruses that selectively replicate in and kill cancer cells [2,3]. There are several reasons why OVT is considered an elegant option for cancer treatment. Firstly, oncolytic viruses exhibit specific replication in cancer cells, often dependent on the genetic changes that discriminate cancer cells from non-malignant cells [4,5]. Hence, genetic changes underlying malignant cell growth are utilized for selective therapy. Secondly, oncolytic viruses kill their host cell as an integral part of their life cycle. Replication in a host cell inevitably leads to death of the host cell. Oncolytic viruses, thus, have profound direct anticancer cell activity. Thirdly, selective replication in cancer cells produces large numbers of progeny viruses that upon their release are capable of infecting new cancer cells causing subsequent cycles of lytic replication. Thus, the anticancer effect is amplified in situ. On top of these direct anticancer effects, oncolytic viruses engage immune system responses of the host [6,7]. Oncolytic virus replication in a tumor can help override the immune suppressive conditions that exist in the tumor microenvironment [7]. Upon virus infection, an antitumor immune response is induced by the production of immune-stimulatory cytokines by virus-infected cancer cells and the concerted release of endogenous danger-associated molecular patterns, virus-derived pathogen-associated molecular patterns, and tumor-specific or tumor-associated antigens from lysed cancer cells in a pro-inflammatory context. This may induce tumor-specific immunity, also against cancer cells at distant sites not subjected to virus infection [7].

Several different viruses are considered for use in OVT. Some of them advanced to clinical trials. Of these, an oncolytic herpes simplex virus (HSV) was recently approved for treatment of melanoma by the competent authorities in the United States and Europe [8,9] and an oncolytic adenovirus was already approved in 2005 in China for treatment of head and neck cancer in combination with chemotherapy [10]. Generally, oncolytic viruses can be divided into three main groups. The first group contains viruses with a natural propensity to preferentially replicate in cancer cells, while being non-pathogenic in humans. Many of these viruses are of non-human origin. This group comprises autonomous parvovirus, myxoma virus, Newcastle disease virus, and reovirus [11,12,13]. Many of the second group consists of viruses that were attenuated by propagation in vitro. These attenuated strains of, e.g., measles virus or vaccinia viruses were used safely in humans as vaccine vectors and were, therefore, considered safe starting points to build oncolytic viruses. The third group consists of viruses that were genetically engineered by introducing mutations in their genome to ensure selective replication in cancer cells. The latter group includes oncolytic viruses derived from adenovirus, HSV, and vesicular stomatitis virus (VSV), as well as an attenuated and genetically modified strain of poliovirus [11,12,13]. Oncolytic adenoviruses, often also referred to as conditionally replicating adenoviruses (CRAds), represent one of the most extensively researched oncolytic viruses. This review focuses on oncolytic adenoviruses, but the general concepts discussed apply to other oncolytic viruses as well.

### Oncolytic Adenoviruses

Adenoviruses are small, non-enveloped double-stranded DNA viruses (34–36 kb), continuously present in the population and responsible for self-limiting infections in immune-competent individuals, including mild respiratory, conjunctival, and gastrointestinal diseases, or cystitis. Infections in immunocompromised patients are much more severe, including infections in liver (hepatitis), lung (pneumonia), or heart (myocarditis) that can result in death (reviewed by Lion [14]). Adenoviral replication is strongly dependent on expression of the immediate early E1A region. To establish a successful replication in a host cell, the adenovirus seizes control over retinoblastoma protein (pRb) and p53 protein (using E1A and E1B viral proteins), which suspends the host cell cycle in the synthesis (S) phase [15]. 

Non-replicating adenoviruses have been used as gene delivery vectors in cancer gene therapy strategies for many years. Replication-deficient adenovirus vectors lack genes from the E1 region and may also have E3 genes deleted to accommodate newly introduced genes [16]. Although these adenoviruses showed low toxicity and promising results compared to conventional therapy, their efficacy in clinical trials remained low, which minimizes their use. 

Current research focuses mainly on CRAds that are considered much more promising agents for the treatment of cancer. Replication of CRAds is controlled either by regulating mostly E1A gene expression with a tumor-specific promoter or by deletions in the viral genome that require cellular factors for compensation [17]. Examples of the first strategy include human telomerase (hTERT) promoter-driven oncolytic adenoviruses [18,19,20]. The first CRAd to be evaluated for cancer treatment, ONYX-015, was made following the second strategy. This adenovirus variant was designed to selectively replicate in p53-deficient cancer cells by deletion of the *E1B55K* gene [21]. Although ONYX-015 showed cancer cell-selective replication, its efficacy was disappointing [22]. Since then, newer generations CRAds with improved selectivity and potency were developed, including Ad5-Δ24 and ICOVIR-5 [23,24]. Nevertheless, despite very encouraging results from in vitro and animal studies, the anticancer efficacy of CRAds, as well as of other oncolytic viruses, as a single agent in humans is generally modest [25]. Thus, there is a clear need to increase the efficacy of OVT. This could be achieved using more effective delivery methods or by enhancing the potency of CRAds to kill cancer cells or to induce an antitumor immune response. In addition, while most efforts are on improving anticancer treatment efficacy, studies are also undertaken to more stringently control CRAd replication in healthy cells.

## 2. Strategies to Increase the Efficacy of Oncolytic Virus Therapy with CRAds

### 2.1. Achieving More Effective Delivery of Oncolytic Adenovirus to Tumors

Effective OVT with CRAds requires that viruses are delivered to tumors in the human body and that they enter cancer cells to initiate oncolysis. Notably, cancer cells are sometimes resistant to CRAd infection due to low expression of the primary receptor molecule coxsackie-adenovirus receptor (CAR) [26]. Typical neoplasms in which downregulation of CAR expression was observed include prostate, colon, and kidney cancers [27]. Retargeting strategies allow overcoming this obstacle, specifically by diversion of the virus to other cell surface receptors. Strategies that were successfully followed to accomplish this were, e.g., incorporation of a cyclic RGD4C peptide motif in the adenovirus fiber knob to allow entry via α_v_β3 and α_v_β5 integrins [28], pseudotyping the viral capsid with proteins from other serotype adenoviruses or with chimeric capsid proteins [29,30], or expressing bispecific adapter molecules from the CRAd genome targeting virus entry via an alternative cell surface receptor [31]. Generally, these modifications resulted in more effective CRAds with broader applicability in OVT.

The administration route to deliver the virus to tumor cells in the human body poses another challenge. Systemic administration of CRAds was proven quite ineffective since most injected virions are eliminated before they reach their target. Much research is put into the development of methods to chemically modify viral capsids to shield them from sequestration in the liver and inactivation by the immune system [32]. Another interesting approach is to use carrier cells as temporary virus hosts delivering oncolytic viruses, including CRAds, to tumor sites. This Trojan horse concept is very attractive, because it not only hides the virus from the immune system, but also exploits the capacity of cells to extravasate from the circulation and home to tissues [33,34]. However, several major challenges remain, including premature expression of viral proteins in the carrier cell, complicated timing of the delivery, acquired adaptive immunity to carrier cells, or the inability to pass through capillaries, which results in the accumulation in, e.g., lungs, and subsequent release of the virus before delivering it to the tumor [33,35,36]. Moreover, there is a contradiction in delivering a virus with cancer-selective replication properties using a non-malignant carrier cell. At least a single virus lifecycle should be completed in this cell to allow release of infectious progeny virus at the tumor site. This means that either the virus should not be entirely cancer-selective, or the carrier cell should have cancer cell-like properties, such as a deregulation in growth control. Both options may raise safety concerns that need to be addressed.

### 2.2. Improving Oncolytic Adenovirus Specificity by Employing microRNA-Dependent Replication

A novel strategy to make CRAds safer by further limiting their replication in non-malignant cells exploits microRNA (miRNA) technology. Here, a miRNA target sequence is incorporated in the 3′ untranslated region (UTR) of the transcript encoding a viral protein essential for replication, usually E1A. Infection of cells expressing the relevant miRNA results in silencing of the essential protein and, consequently, suppression of viral replication. In recognition of the fact that human adenoviruses have strong tropism for liver cells and are efficiently delivered to liver cells when administered systemically, which could lead to liver toxicity, particular efforts were made to inhibit CRAd replication in liver cells. Ylösmäki et al. [37] constructed Δ24-type CRAds carrying three target sites for the liver-specific miRNA miR-122 in the Δ24-E1A 3′ UTR. While this reduced Δ24-E1A expression in miR-122-expressing hepatoma cells by more than 90%, this did not inhibit virus replication. By introducing a second mutation in the E1A messenger RNA (mRNA) that reduces overall expression levels, this problem was successfully addressed. The resulting virus was severely attenuated, failing to induce cytopathic effects in hepatoma cells and producing more than 10,000-fold reduced titers in hepatoma cells compared to non-small lung cancer cells. The same approach was taken independently by others; CRAds with miR-122 binding sites exhibited reduced E1A expression and virus replication in liver cells and almost absent liver toxicity upon injection in mice [38,39].

Knowledge of differential expression of miRNAs in cancer cells versus non-malignant cells was used similarly to construct CRAds with reduced replication in non-cancer cells. In particular, miR-199, which is downregulated in cancer cells, was considered for this purpose [19,40]. Significantly reduced virus replication and toxicity was observed in non-malignant cell cultures and in livers of newborn mice if miR-199 target sequences were incorporated into the E1A 3′ UTR. By combining target sequences for miR-199a and liver-specific miR-122a in the 3′ UTR of an hTERT promoter-driven E1A gene, a most selective CRAd was obtained [19]. Oncolytic potency in cancer cell lines was fully retained. Others translated the general concept of miRNA-dependent suppression of CRAd replication to more specific applications, where viral toxicity might arise upon local tissue administration. MiRNA-regulated CRAds were made in a wild-type adenovirus background for selective treatment of pancreatic cancer and glioma [41,42]. The viruses were effective in animal models and exhibited strongly reduced toxicity profiles. For this, insertion of four miRNA target sites appeared sufficient, but this might depend on host miRNA expression levels and the strength of the viral gene promoter. Thus, preclinical data suggest that host cell miRNA-regulation of adenovirus replication alone can already provide sufficient CRAd selectivity. However, further development of this type of CRAd will likely be done in the background of CRAds carrying mutations with proven clinical safety.

Another strategy to restrict CRAd replication in healthy cells was taken by Gürlevik et al. [43]. They expressed multiple artificial miRNAs targeting essential adenovirus genes from a single polycistronic transcript, driven by a tumor suppressor p53-dependent promoter, inserted in the adenovirus genome. In cells with functional p53 expression, but not in p53 mutant cells, the artificial miRNAs were expressed and silenced their target genes. Although this attenuated viral replication in p53 wild-type cells, virus progeny production was not prevented. The p53-dependent attenuated phenotype was also observed in mice injected with the virus, where p53 wild-type mice carried lower virus copy numbers in their liver than did p53 knockout mice. The inhibition of virus replication that could be achieved in this study appeared inferior to that achieved when relying on host miRNAs. This can possibly be explained by the fact that the latter are already highly expressed in the host cell at the time of infection, whereas CRAd-encoded miRNAs need to be produced after cell entry and their level of expression might be replication-dependent, as copy numbers increase tremendously during replication. Hence, functional knockdown of viral genes by virus-encoded miRNAs is probably not reached before viral replication is initiated.

### 2.3. Strategies for Improving the Potency of Oncolytic Adenoviruses

The strong selectivity of CRAd replication in cancer cells that is obtained with currently available technology, providing excellent safety, allows exploration of methods to enhance the cancer cell-killing potency of CRAds. Apart from the tropism modifications mentioned above, investigated approaches include the introduction of potency-enhancing genome mutations and the expression of potency-enhancing transgenes. The former is usually based on identification of such mutations in virus variants that are either naturally occurring or were selected using molecular evolution methods. A good example of this approach is the genetic bioselection of a virus with a truncating mutation in the endoplasmatic retention domain of the E3/19K protein that promotes virus progeny release from infected cells [44]. Incorporation of this T1 mutation in an Ad5-Δ24 backbone with RGD4C fiber modification produced a CRAd with strong antitumor activity in vitro and in vivo, without compromising its selectivity toward cancer cells [45]. Another example is ColoAd1, a recombination product between two naturally occurring serotypes selected for rapid replication on human cancer cell lines, which exhibits increased potency in particular on colorectal carcinoma cells [46]. Transgenes that can be carried by the virus to increase its potency include genes encoding fusogenic proteins or tissue-remodeling peptides, enzymes for use in suicide gene therapy approaches, pro-apoptotic or tumor suppressor proteins, and immune-modulating proteins. Expression of a fusogenic protein induces cell–cell fusions, forming syncytia consisting of CRAd-infected cells and adjacent non-infected cells, leading to death also of the fused non-infected cells and, thus, more effective tumor eradication [47]. Another way to promote spread of CRAds through tumor tissues is to express molecules that remodel the extracellular matrix, allowing better virus dispersion in a solid tumor mass [48,49,50]. The suicide gene therapy concept, where an enzyme is expressed that converts a non-toxic pro-drug into a toxic compound, was also applied in the context of CRAds [51,52]. This was met with various levels of success, as premature death of the host cell inhibits virus propagation. Careful timing of virus and pro-drug administration is, thus, essential, which makes the design of clinical treatment schedules challenging. The same can be said about expression of apoptosis-inducing proteins, where premature cell death inhibits, and properly timed cell death promotes virus spread [53]. In contrast, expression of tumor suppressor protein p53 was not met with this limitation, presumably because adenovirus-encoded E1B55K protein interacts with p53 and regulates its activity during adenovirus replication [54,55]. In the late phase of the virus lifecycle, p53 promotes cell death, thereby accelerating virus progeny release [54,56]. In addition, p53 increases late adenoviral gene expression, presumably by enhancing transcription from the virus major late promoter (MLP) [57] (Figure 1). Exogenous p53 expression augmented CRAd replication in the majority of cancer cell lines, primary tumor cell cultures, and xenograft tumor models tested [54,55,58,59,60,61,62,63]. Notably, transgene expression from the adenovirus genome during virus replication in host cells can be delayed by coupling its transcription to the MLP. Apart from avoiding premature transgene expression, this strategy offers the additional advantage of confining transgene expression to cancer cells in which the CRAd replicates selectively [64]. In addition to the aforementioned ways of enhancing the primary tumor-eradicating potency of CRAds, there is currently much interest in promoting the secondary, immune-stimulating effect of these viruses, by incorporating genes encoding immune-stimulatory molecules, such as cytokines, chemokines, and checkpoint inhibitors (reviewed by de Gruijl et al. [7]). Altogether, these developments led to substantially improved CRAd variants with more potent cancer cell-killing properties than earlier generation viruses. Nevertheless, one area of potential CRAd improvement was long overlooked. While cancers may lack certain molecules needed for effective OVT that can be supplemented by transgene expression from the CRAd genome, they could also express molecules inhibiting OVT that should, thus, be depleted. Currently available gene suppression techniques allow investigating this and constructing even more powerful CRAds capable of overcoming obstacles for OVT in cancer cells. The next section focuses on this new development.

## 3. Gene Suppression to Make Oncolytic Viruses More Effective

There are two requirements for improving OVT by suppressing inhibitors of CRAds in cancer cells, i.e., (i) identification of the inhibitory gene products in cancer cells, and (ii) design of an effective way of suppressing genes, using molecules that can be carried by CRAds. For the former, one may build on available knowledge of virus–host cell interactions or use high-throughput functional screening approaches. The latter, we know now, can be successfully achieved by using RNA interference (RNAi).

Currently, RNAi is the most widely used method to specifically downregulate functional gene expression. RNAi is a conserved cellular surveillance system in eukaryotes that activates a sequence-specific degradation of RNA species homologous to short non-coding double-stranded RNA (dsRNA) molecules [65]. These dsRNA molecules, including miRNAs and short interfering RNAs (siRNAs), are products of the RNase III enzymes Drosha and Dicer [66]. One of the strands of the miRNAs and siRNAs, the active or leading strand, is incorporated into a multiprotein complex called RNA-induced silencing complex (RISC) and directs the target RNA recognition leading to its degradation. In addition, miRNAs direct translational repression and mRNA deadenylation, further contributing to gene silencing [67]. The mechanism of this strongly conserved regulation of cellular mRNAs by miRNAs was thoroughly reviewed in several publications [65,68,69].

A means of producing exogenous siRNAs or miRNAs in a cell is to express small RNA molecules as single transcripts that form a stem-loop structure, generally referred to as short hairpin RNAs (shRNAs) and short hairpin miRNAs (shmiRNAs). Their structure is very similar to that of pre-miRNA, i.e., the Drosha-cleaved product in endogenous miRNA biogenesis. The dsRNA in the stem of these molecules is processed by Dicer to form mature siRNA and miRNA duplexes, respectively, that will then incorporate their active strand into RISC and direct RNAi. Successful RNAi induced by adenovirus-encoded exogenous shRNA was achieved initially using replication-defective vectors [70]. This opened the possibility of silencing genes in cancer cells with a therapeutic intent. 

Because the efficiency of gene delivery with replication-defective vectors is generally considered inadequate for effective cancer treatment, interest rose in using CRAds as shRNA delivery vectors. However, achieving effective RNAi in cells infected with a replication-competent adenovirus was not considered trivial. RNAi is recognized as a cellular defense mechanism against virus infection and this led many viruses to evolve molecules that inhibit RNAi [71,72]. Although the importance of RNAi as an antiviral response in mammalian cells is heavily debated (e.g., References [73,74]), there was reason to assume that the process of RNAi would be hampered in cells in which a virus is replicating. In fact, several independent reports showed that adenovirus-encoded virus-associated RNAs (VA RNAs) inhibit the RNAi machinery (Figure 2). VA RNAs (I and II) were found to inhibit the host cell RNAi pathway at the level of nuclear export, by competing with miRNA precursor molecules for binding to Exportin 5; at the level of processing RNAi precursor molecules, by inhibiting the activity of Dicer; and at the effector level, by competing with siRNA and miRNA for incorporation into RISC [75,76]. In addition, it was later reported that the titration of Exportin 5 by VA RNA also reduced Dicer mRNA nuclear export and, thereby, Dicer protein levels [77]. Apart from this direct inhibition of the RNAi machinery by virus-encoded RNA, it was recently reported that the type I interferon response evoked by adenovirus vectors in cells inhibits the processing of shRNAs by Dicer, resulting in a less effective knockdown of target genes [78]. Altogether, there were reasons to expect that effective CRAd replication and RNAi could be incompatible.

Nevertheless, in a proof-of-concept study using firefly luciferase as target, Carette et al. showed that CRAd-induced shRNA can induce proper gene silencing in human cancer cells [79]. That suggested that, although interferon-responses and VA RNAs inhibit the RNAi machinery, they do not prohibit RNAi brought about by adenovirus-encoded shRNA. Still, Machitani et al. showed that RNAi mediated by adenovirus-encoded shRNA can be improved by deleting VA RNA sequences [80]. Notably, this study was done with replication-defective adenovirus vectors. Since VA RNA expression is highly replication-dependent, competitive incorporation into the RNAi machinery is expectedly even higher in a replication competent context. Thus, to achieve highly effective RNAi using CRAds it could be considered to (partly) delete VA RNA sequences. However, adenovirus lacking major VA RNA I cannot efficiently translate viral mRNA in the late phase of the infection, resulting in poor replication [81]. In addition, it was recently reported that VA RNA II-encoded viral miRNA promotes adenovirus replication, presumably via suppression of cullin 4-mediated inhibition of Jun N-terminal kinase (JNK) signaling [82]. VA RNA-depleted CRAds are, therefore, considered less attractive agents for OVT. However, observations reported by Kamel et al. suggested that VA RNA I expression is required primarily to counteract cellular anti-viral interferon responses, not so much the RNAi machinery [83]. This notion is supported by the observation that processing of VA RNAs by Dicer into viral miRNAs inhibits adenovirus replication, apparently by depleting VA RNAs capable of inhibiting double-stranded RNA-activated kinase (PKR) [84]. Thus, the RNAi machinery appears to contribute to the cellular defense against adenovirus infection indirectly, by inhibiting the capacity of the virus to suppress the major anti-viral interferon–PKR response. This implies that if the PKR and RNAi inhibitory functions of VA RNA I can be dissected by introducing specific mutations, or, alternatively, VA RNA I is deleted and the interferon response is inhibited through other means, such as an shRNA targeting PKR, more effectively replicating and gene-silencing CRAds could perhaps be made.

Although the efficacy of CRAd-mediated RNAi can probably be improved, the findings by Carette et al. [79] fueled the development of CRAds capable of silencing a variety of target genes in cancer cells to achieve more effective anticancer treatment. In many cases, the target genes were not necessarily expected to impact the efficacy of OVT, but were chosen merely for their known or anticipated anticancer effects. Examples include, among others, silencing an oncogene to inhibit tumorigenicity, silencing a pro-angiogenic growth factor to inhibit angiogenesis, or silencing an apoptosis inhibitor to promote cell death [85,86,87]. These RNAi-inducing CRAds could, thus, be considered therapeutic agents providing a combination of OVT and targeted therapy. The two independent activities provided by CRAd RNAi were often additive, resulting in a more effective treatment in preclinical models than could be achieved with unarmed control CRAds or replication-defective RNAi vectors. Below, we focus on approaches to achieve more effective OVT.

### 3.1. Combining OVT with Suppression of CRAd-Inhibitory Target Genes in Cancer Cells

As a logical extension of their above-described observation that Dicer inhibits adenovirus replication, Machitani et al. developed a CRAd expressing an shRNA against Dicer [88]. While replicating in cancer cells, the virus silenced Dicer expression, allowing an increase in VA RNA copy numbers available for inhibiting PKR activity. Consequently, the virus produced higher genome copy numbers and functional progeny titers than did a control virus silencing luciferase. This resulted in a more effective lysis of cancer cells in vitro and a more effective inhibition of tumor growth in vivo. Apparently, in this respect, the gain of more effective VA RNA I-mediated PKR inhibition was more important than the loss of VA RNA II-mediated JNK activation. Although expression of the shRNA was driven by the H1 mammalian polymerase (Pol) III promoter, which is not expected to provide cancer cell selectivity, Dicer expression was not significantly reduced in non-malignant cells. This can best be explained by the fact that CRAds do not replicate in non-cancer cells and, consequently, genome copy numbers in these cells remain very low. Thus, while it can be argued that inhibition of the RNAi machinery in cells could interfere with endogenous miRNA processing causing unpredictable and possibly detrimental effects, the efficiency of shRNA expression in the absence of viral replication was apparently low enough to avoid such unwanted effects.

Another approach that was successfully taken to increase the potency of CRAds is to suppress cell-cycle inhibition in cancer cells. Since CRAds, by design, lost their capacity to take control over the cell cycle, to induce the S phase for efficient genome replication, their replication efficiency depends on the spontaneous progression of cancer cells through the cell cycle. Although proper regulation of the cell cycle is lost in cancer cells, partial regulation of cell-cycle progression may still exist. It could, therefore, be conceived that cell-cycle inhibition, in particular at the Gap 1 (G1) checkpoint, inhibits adenovirus replication. This notion was supported by the finding that overexpression of the major cyclin-dependent kinase (CDK) inhibitor p21^WAF1/CIP1^ in cancer cells inhibited CRAd replication [89] and by the observation that CRAds replicated more efficiently, inducing stronger cytopathic effects, in p21^WAF1/CIP1^ knockout cells [90]. In experiments where CRAds were combined with siRNAs targeting the CDK inhibitors p21^WAF1/CIP1^ or p27^KIP^, virus progeny production and cell-killing potency were increased, the latter depending on cellular p53 status [90] (Figure 1). These observations provided incentive to construct a prostate-specific CRAd expressing an shRNA targeting p21^WAF1/CIP1^ [91]. As expected, the virus exhibited increased replication in and oncolysis of prostate cancer cells in vitro and provided enhanced tumor suppression and survival in a tumor xenograft animal model. Interestingly, apart from the sought effect on adenovirus replication via cell-cycle control, a less anticipated positive effect was observed as well. Silencing p21^WAF1/CIP1^ increased expression of the androgen receptor in prostate cancer cells, thereby stimulating expression of the viral E1A gene driven by the used androgen receptor-dependent prostate-specific promoter. While the relative contributions of both effects are difficult to dissect, the total effect of expressing a p21^WAF1/CIP1^-silencing shRNA from the CRAd genome on the efficacy of OVT was evident.

A foreseeable further improvement of CRAd potency could be to arm the CRAd with both a p53 gene and a p21^WAF1/CIP1^-silencing shRNA. As discussed above, the utility of expressing p53 from the CRAd genome to increase its potency was shown in a variety of cancer models. Since p21^WAF1/CIP1^ is a direct transcriptional target of p53, an unwanted side effect of expressing p53 could be induction of p21^WAF1/CIP1^ expression, limiting CRAd replication. Combined expression of p53 and an shRNA targeting p21^WAF1/CIP1^ could then overcome this limitation. A replication-defective adenovirus vector with cocistronic expression of p53 and a p21^WAF1/CIP1^-targeting artificial miRNA was already tested for its anticancer properties [92]. This viral vector caused more effective apoptosis induction in cancer cells and decreased xenograft tumor growth compared to control vectors expressing only p53 or p21^WAF1/CIP1^-targeting miRNA. A similar favorable combination effect of expressing p53 and silencing p21^WAF1/CIP1^ is to be expected in the context of OVT with a CRAd and, therefore, deserves to be explored.

Along the same line of reasoning, CRAds expressing p53 could probably be made more effective by silencing p53 inhibitors (Figure 1). Many p53 inhibitors are known and confirmed active in cancer cells. Paradigm examples are human papillomavirus (HPV)-E6 protein in cervical cancer cells and amplified mouse double minute 2 homolog (MDM2) in sarcomas. We already showed that CRAds expressing p53 variants incapable of binding to HPV-E6 or MDM2 are more effective in these cancers [61,62]. A limitation of this approach could be that the introduction of mutations in the p53 protein to abrogate binding to its inhibitor might compromise its transcriptional activity. Therefore, combining the expression of wild-type p53 with an shRNA targeting the inhibitor could perhaps be more effective. In addition, cancer cells often express multiple p53 inhibitors. This can be deduced from functional screens for p53 modulators done on cancer cell lines [93,94,95]. Preventing binding to a single inhibitor might, therefore, not be sufficient in many cases. It will be difficult, if not impossible, to prevent binding of p53 to multiple inhibitors by mutating its amino-acid sequence. In contrast, since their small size allows for multiple shRNAs to be inserted into a single CRAd genome, it should not be too difficult to simultaneously silence multiple p53 inhibitors, maximizing the activity of p53-expressing CRAds.

### 3.2. Combining OVT with Targeting Immune Suppression

As mentioned above, stimulation of the immune system to evoke an anti-tumor response is currently considered a crucial property of oncolytic viruses. This inherent property can be further enhanced by arming the virus with, e.g., immunostimulatory cytokines [7]. While this resulted in much improved OVT systems, active immune suppression in the tumor microenvironment was not fully addressed. Many immune-suppressive molecules, expressed by cancer cells or by suppressive cells of the immune system, were already identified in the tumor microenvironment. Surprisingly, with the technology to combine CRAds with RNAi at hand, arming CRAds with shRNAs targeting such molecules is still quite an unexplored approach. The exception is a CRAd encoding a combination of MART1 melanoma antigen, granulocyte-macrophage colony-stimulating factor (GM-CSF) cytokine, and an shRNA silencing transforming growth factor β2 (TGF-β2) [96]. The rationale for the combination was that, while GM-CSF has many immune-stimulating properties, such as inducing maturation of dendritic cells (DCs), expansion and differentiation of lymphocytes, and recruitment of natural killer (NK) cells, it is also known to enhance the expansion of myeloid-derived suppressor cells (MDSC). TGF-β2 antagonizes GM-CSF-induced DC maturation, inhibits tumor antigen-specific T-cell activation, and stimulates MDSC. Concomitant expression of GM-CSF and suppression of TGF-β2 was, thus, expected to provide a more potent induction of antitumor immunity than expressing GM-CSF alone. This CRAd was tested in a prime-boost therapeutic vaccination strategy, using immune-competent mice and mouse melanoma cells engineered to allow human adenovirus replication. A plasmid expressing MART1 was injected intramuscularly to generate MART1-specific memory T cells and the CRAd was injected intratumorally to boost the immune response. The strategy significantly increased tumor infiltration with cytotoxic T cells, NK cells, NKT cells, (mature) DCs, and macrophages, and decreased regulatory T cells. Immune cell activity was evident from high levels of interferon-γ (IFN-γ) expressed. Together with the oncolytic activity of the virus, this resulted in delayed tumor growth. The contribution of silencing TGF-β2 to the total effect was, however, not entirely clear from the reported results.

Another CRAd that could reactivate antitumor immunity through suppressing an immune-suppressive target gene in cancer cells was made, but not tested as such. This CRAd expressing antisense RNA against signal transducer and activator of transcription 3 (STAT3) was shown capable of depleting STAT3 protein in infected cancer cells, as well as its downstream targets [97]. The virus exhibited increased replication and more potent antitumor activity, presumably through counteracting the effects of STAT3 on cell survival, cell migration, and angiogenesis. STAT3 is, however, also involved in immune evasion. It would, therefore, be very interesting to investigate the effects of the STAT3-suppressing virus on the antitumor immune response. The host specificity of human adenoviruses is, however, a major challenge in the design of such experiments. It is very difficult to develop an immune-competent animal model with available tools for immunological studies, that allows replication of human adenoviruses. Although Syrian hamsters are semi-permissive for human adenovirus replication and, thus, provide a useful immune-competent animal model for CRAd efficacy studies [98], and CRAds armed with immune-stimulating transgenes were successfully tested in these animals (e.g., References [99,100]), only few antibodies are available to study hamster immune cell subsets, precluding in-depth analysis of immune responses. For this, ex vivo cultures of human whole tumor-derived single-cell suspensions or fresh tissue slides, or novel mouse models carrying patient-derived xenograft tumors and reconstituted with immune cells from the same donor could prove highly valuable.

### 3.3. Exploiting Virus–Host Interactions via MicroRNAs

Although, as mentioned above, a physiological role for RNAi in response to virus infection in mammals is being challenged, miRNA-mediated virus–host interactions were described for a number of mammalian viruses (reviewed in References [101,102,103]). This includes anti-viral or pro-viral effects of host miRNAs, as well as virus-encoded miRNAs that modify host cell biology to establish an environment conducive to completion of the virus lifecycle or that exploit the host RNAi machinery to regulate viral gene expression. Paradigm examples of human miRNAs that promote virus replication include miR-122 and miR-132 that promote hepatitis C virus replication and human immunodeficiency virus 1 (HIV-1) replication, respectively [104,105]. Inhibitory effects of human miRNAs on viruses are, however, more common. They could be part of the host innate antiviral response to recover from an infection, or alternatively, be part of a strategy of the virus to establish a persistent infection [106].

In the case of adenovirus, the VA RNAs constitute the virus-encoded miRNAs (Figure 2). They are processed by the RNAi machinery and are functional in directing silencing of target sequences [76,107,108]. As mentioned above, VA RNA II-encoded viral miRNA was shown to promote adenovirus replication [82]. The functional importance of VA RNA I acting as miRNAs is, however, questioned. While specific inhibition of miRNA target site binding using chemically modified oligonucleotides (i.e., antagomirs) reduced adenovirus replication [76], introduction of mutations in the miRNA seed sequence did not appear to influence adenovirus propagation [83]. Nevertheless, specific cellular target genes for VA RNA I-encoded miRNA, including T-cell-restricted intracellular antigen 1 (TIA-1), were identified and their silencing was confirmed in virus-infected cells [109]. The relevance of silencing these genes for the virus lifecycle is so far mostly unresolved. However, it seems unlikely that adenoviruses evolved specific silencing of target genes in host cells without any functional importance.

Adenoviruses also modulate the expression of host miRNAs. We found that, on top of a general suppression of miRNA levels in adenovirus-infected cancer cells, which is in agreement with the described interferon- and VA RNA-dependent effects on the RNAi machinery, adenovirus serotype 5 significantly reduced expression of a subset of human miRNAs, in particular miR-27, miR-100, miR-155, miR-181, and miR-222 in the late phase of replication [110]. Interestingly, others reported that these miRNAs were also downregulated in cells infected with other viruses, suggesting that these miRNAs could be involved in general antiviral responses, and that viruses evolved specific countermeasures against them. In contrast, only very few miRNAs were induced upon adenovirus infection [110]. Interestingly, among these was miR-132, which was also reported overexpressed in human monocytes infected with several other viruses, suppressing innate antiviral immunity and promoting viral replication [111]. Hence, different viruses exhibit striking similarities in their communication with host cells via miRNAs. As miRNAs usually have many targets, modulation of a single host miRNA by a virus can have complex biological effects, silencing multiple genes that together impact virus replication. In addition, not all target genes predicted on the basis of sequence complementarity are validated as genuine miRNA targets. They may not or only under certain circumstances be silenced. This makes it very difficult to dissect virus–host interactions via miRNAs and identify the crucial host genes contributing to virus biology.

Since miRNA expression is deregulated in cancer cells, CRAds that are replicating in cancer cells might encounter miRNA expression patterns that differ from the ones that human adenoviruses evolved to cope with in their natural differentiated host cells. Consequently, CRAds might not exploit host miRNA expression to their full benefit, and host antiviral miRNA responses to viral infection might not always be effectively counteracted. It can be envisioned, therefore, that overexpression or inhibition of certain miRNAs in cancer cells increases CRAd replication and, thus, oncolytic potency. The identification of miRNAs or miRNA inhibitors for this purpose could perhaps be based on known virus–host interactions and deregulated miRNA expression patterns. For example, tumor suppressor miR-34a expression is often absent or low in cancer cells as a consequence of chromosomal deletion or p53 deficiency. As a direct transcriptional target of p53 and silencer of genes required for cell proliferation and survival, miR-34 mimics part of the p53 activities [112]. In view of the established effects of expressing p53 on CRAd efficacy, it could, thus, be envisioned that expressing miR-34a from the CRAd genome similarly promotes CRAd propagation and cancer cell killing (see Figure 1). MiR-34a-expressing CRAds were presented in two studies. In the first study, hTERT promoter-driven E1A-Δ24-type CRAds were armed with miR-34a and/or interleukin-24 (IL-24) and tested in hepatocellular carcinoma models [20]. The cytokine IL-24 was chosen for its pro-apoptotic and anti-angiogenic properties. Knockdown of miR-34a target genes was confirmed in infected cells. The effects of miR-34a on cytotoxicity in vitro and tumor growth inhibition in vivo were minimal. In contrast, the effects of expressing IL-24 were significant and most prominent anti-tumor effects, including complete tumor regression, were achieved with the virus carrying both miR-34a and IL-24. This suggests that the augmented efficacy of the CRAds depended primarily on the anti-tumor activities of IL-24. In the second study, carcinoembryonic antigen promoter-driven E1A-type CRAds were armed with miR-34a and/or miR-126 and tested in pancreatic cancer models [113]. These two tumor suppressor miRNAs were chosen because of their low expression in pancreatic cancer and confirmed anti-tumor effects. Also in this study the effect of expressing miR-34a was very small. The significance of miR-34a expression for tumor growth control in comparison to a control CRAd was not demonstrated. Interestingly, the miR-34a-expressing CRAd appeared to produce somewhat lower virus titers in pancreatic cancer cell lines than the control virus. In this respect, the virus, thus, did not reproduce previous observations made with p53-expressing CRAds.

Apart from rationally choosing miRNAs to empower CRAds, one may also take an unbiased approach to identify useful miRNAs through functional screening. The availability of genome-wide miRNA mimic and inhibitor libraries allows global analysis of miRNA effects on the lifecycle of viruses [110,114]. Screening for miRNA effects on the replication of a number of human and mouse herpes viruses, including human and mouse cytomegalovirus (CMV), as well as validation on an evolutionary unrelated RNA virus, identified several miRNAs with broad proviral or antiviral properties [114]. In particular, miR-30 broadly stimulated virus replication, and miR-199a and miR-214 inhibited virus replication. Since these observations were made on a diversity of viruses, the miRNAs are not likely to act on viral elements. Instead, most probably, they regulate host cell pathways that support or protect against virus replication. Interestingly, miR-199a and miR-214 are expressed from the same intronic cluster [115] and were found downregulated in CMV-infected cells. Thus, these viruses appear to counteract two antiviral miRNAs by suppressing a single transcript. While miR-30 was not reported differentially expressed in this study, several other viruses were shown to induce the expression of miR-30 variants [116,117]. Intriguingly, on one hand, this upregulation appears IFN-dependent [116], whereas, on the other hand, miR-30 inhibited type-I IFN signaling, allowing increased virus replication [117]. In any event, these findings suggest that certain virus–host interactions via miRNAs are broadly conserved. Hence, observations made in miRNA screens with one virus might be applicable to other viruses as well.

We performed screens for adenovirus replication and cytotoxicity on prostate cancer cells using the same miRNA libraries as were used in the aforementioned study [110]. We identified several miRNAs that stimulate adenovirus-induced cell death. Some of these inhibited adenovirus replication, possibly as a consequence of premature cell death occurring before adenovirus replication was completed. In general, strong induction of cell death was associated with reduced virus production. The exception was miR-181a. Expression of this miRNA strongly induced cell death and slightly increased virus production. The most striking results were obtained with miR-26b. Transfection of miR-26b mimics increased adenovirus-induced cell death, as well as infectious virus production and release. Consequently, adenovirus propagated more rapidly in human prostate cancer cell lines, producing larger plaque sizes. This makes miR-26 a promising candidate for incorporation in the genome of CRAds, to augment OVT of prostate cancer and perhaps other cancers. Construction and characterization of such a novel CRAd is, therefore, warranted. An important question that needs to be answered then is if the timing and levels of miRNA expression obtained during CRAd replication in cancer cells will reproduce the functional effects obtained by combining miRNA mimic transfection and virus infection.

### 3.4. RNAi Screening for Inhibitors of Oncolytic Virus Efficacy in Cancer Cells

Clearly, virus–host interactions are complex and our current knowledge of the cellular pathways involved is incomplete. There are, therefore, probably many more opportunities to improve the potency of oncolytic viruses than we can rationally hypothesize. The advent of high-throughput screening technology with whole-genome RNAi libraries offered the opportunity to functionally interrogate every single gene, one by one, for its biological role. Many investigations used this technology to identify cellular factors involved in susceptibility to virus infection (reviewed in Reference [118]). These screens helped identify genes responsible for viral entry, uncoating of the virus, replication of the genome, particle assembly, and spread of progeny viruses. Not surprisingly, these studies focused primarily on viruses that constitute a threat to public health, where the knowledge obtained can be used to develop novel antiviral treatments. However, genome-wide RNAi screening can also be used to identify determinants of oncolytic virus anticancer efficacy. With the ultimate goal of increasing the efficacy of OVT by arming the virus with an shRNA in mind, the most important factors of the virus–host interaction to be studied are those involved in virus genome replication and infectious progeny production, host cell killing, and modulation of the immune response. Early steps in the virus lifecycle, such as entry, uncoating, and early gene expression are of less relevance, since these processes cannot be influenced by virus-encoded shRNAs.

Functional genomic screens usually have a single readout of biological effect. Although high-content (image) analyses can measure more complex phenotypes, such as multiple descriptors of cellular morphology, most screens measure a single parameter, such as cell viability, reporter gene expression, or binding/uptake/secretion of a (fluorescent) molecule. Hence, the most appropriate readout for oncolytic virus efficacy needs to be carefully chosen. In addition, it is crucial that the screening effort is followed by thorough examination of the effects of hits on other aspects of the virus lifecycle that are not measured in the discovery screen. For example, in our miRNA library screens described above [110], we identified a number of miRNAs that strongly promoted adenovirus-induced cell death, but this was at the expense of reducing infectious progeny virus production. Thus, interference with host cell gene expression can have opposing effects on different steps in the virus lifecycle, negating the overall effect of gene silencing on OVT. 

Until now, there are a few examples reported to show that genome-wide RNAi screening can be used effectively to increase our understanding of the interactions between oncolytic viruses and their host cancer cells and to identify determinants of their anticancer efficacy [119,120,121].

In the first example, an arrayed siRNA screen was performed, measuring effects on the killing of three cancer cell lines by Maraba virus [119]. Although Maraba is not a widely used oncolytic virus, it is related to the more commonly used VSV, and the major findings in this study could be validated for both viruses. Among the hits identified in the screens, genes within endoplasmic reticulum (ER) stress response pathways were enriched. Knocking down a key gene in the ER stress response pathway activated caspase-dependent apoptosis in virus-infected cells. This increased oncolytic virus potency in many cancer cell lines, reaching up to four orders of magnitude more effective killing. Importantly, augmented killing was much less pronounced on non-malignant cells, providing a high therapeutic index. Notably, in the screen design used, siRNAs were transfected three days before virus infection. Thus, ER stress responses were already inhibited at the time of infection. In follow-up time scheduling experiments with chemical compounds targeting key proteins in the ER stress response pathway, it was found that virus-induced cell killing was augmented when ER stress responses were inhibited prior to infection, but not when they were inhibited during infection. This pointed at a preconditioning process, where inhibition of ER stress response rewired cellular signaling, leading to increased cell death upon subsequent virus infection. Hence, while the findings of this study can be used to develop combination treatments consisting of preconditioning with a chemical compound followed by virus administration, they do not provide footholds for arming viruses with transgenes or shRNAs. For this purpose, it is probably better to delay gene silencing relative to virus infection in the screen design.

In the second example, a pooled shRNA cell depletion screen was done on breast cancer cells that were infected with an oncolytic HSV [120]. This identified a component of the RNA splicing machinery, arginine-rich splicing factor 2 (SRSF2). Although the effects of silencing SRSF2 were not very strong, they could be reproduced on different cell lines and with different reagents. Silencing SRSF2 changed the abundance of pro- and anti-apoptotic mRNAs, and increased apoptosis. In contrast, it did not stimulate virus replication. In fact, a trend toward inhibition of infectious virus production was observed. SRSF2 depletion appeared to dampen HSV-induced activation of mammalian target of rapamycin (mTOR) signaling, thereby probably inhibiting multiple processes, including cell cycle, virus replication, and cellular antiviral responses. Thus, the effects of targeting SRSF2 could promote, as well as inhibit, the HSV oncolytic potency. Apparently, under the applied experimental conditions the overall effect was more potent cytotoxicity. Importantly, also in this study, gene knockdown or knockout was already established before cells were subjected to the virus. It is difficult to predict if delayed silencing of SRSF2 after virus entry will also tip the scales toward increased oncolysis. Interestingly, treatment of cancer cells with an mTOR inhibitor increased the oncolytic potency of CRAds [122,123]. This suggests a similarity in the virus–host interactions of different oncolytic viruses that could allow translation of the findings to broader applications.

The third example is the systematic analysis of human cell factors affecting replication of Myxoma virus [121]. Genome-wide arrayed siRNA screens were done on a breast cancer cell line that was infected with virus three days after siRNA transfection. A recombinant virus was used that expresses a marker gene in the late phase of replication. Marker gene expression served as a surrogate marker for virus replication. The screens yielded a large database of genes that either enhance or inhibit Myxoma virus replication. Follow-up experiments zooming in on certain cellular pathways provided support for their role in virus replication, but interpretation was difficult and some observations seemed contradictory. This is illustrated, e.g., by the observations made when silencing the Raf/mitogen-activated protein kinase kinase (MEK)/extracellular signal-regulated kinase (ERK) pathway. Whereas silencing of many genes upstream of MEK/ERK enhanced virus replication, silencing more downstream pathway genes, including ERK1, inhibited virus replication. Effects of targeting single genes were usually modest. This is in agreement with the concept that viruses modulate cellular pathways by converting multiple switches in a concerted manner, and that host cells respond to virus infection by activating many systems that collectively create antiviral defenses. Finally, based on the known involvement of RNA helicases in permissiveness of cells for replication of a variety of viruses, Rahman et al. [124] performed a focused siRNA library screen on cervical cancer cells using marker gene-expressing Myxoma viruses, followed by confirmation experiments in other cell lines. This effort identified several RNA helicases that consistently increased or decreased Myxoma virus production. An intriguing finding was that knockdown of retinoic acid-induced gene 1 (RIG-1) reduced Myxoma replication. This was highly unexpected, because of the known role of RIG-1 in antiviral innate immune responses. Knockdown of five RNA helicases stimulated Myxoma virus production, without an apparent effect on viral gene expression or replication. Although their mechanism of action remains to be resolved, these genes could be appropriate targets to construct more potent oncolytic Myxoma viruses.

Similar screens, as far as we are aware, are yet to be reported for CRAds. In addition, despite the recognized crucial role of the immune system in the anticancer effects of oncolytic viruses, RNAi screens with oncolytic viruses have not studied this important aspect. While the design of relevant high-throughput assays for modulation of the immune system will be a challenge, such efforts are warranted. They are expected to provide highly valuable information to design next-generation CRAds for more effective OVT.

## 4. Conclusions

For the past decades, molecular cancer and virology research advanced forward with incredible speed and efficiency. This brought OVT from the bench to the bedside. Novel molecular targets were identified, oncolytic virus constructs designed, and combined therapies proposed or tested in clinical trials. So far, only a few OVTs obtained market authorization, but others are likely to follow. Oncolytic virus therapy moved from a mere concept to a functioning approach. RNA interference became one of the successfully applied methods to make OVT more effective. Here, knowledge of miRNA expression in cancer and healthy cells, as well as of miRNA-mediated virus–host interactions, is implemented. Clearly, a great amount of work needs to be done before OVT will reach its full potential. The contribution of RNAi technology to reaching this goal is difficult to predict. However, although the battle is not over, new research results are constantly made that bring us closer to victory.

## Figures and Tables

**Figure 1 cells-07-00228-f001:**
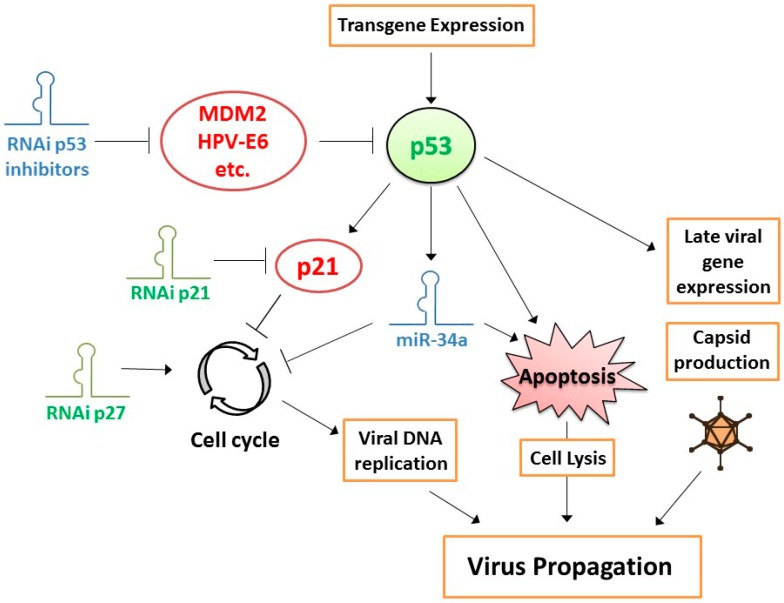
Tumor suppressor p53-dependent strategies to increase the potency of conditionally replicating adenoviruses (CRAds). Expression of functional p53 in cancer cells promotes CRAd propagation by stimulating late viral gene expression and accelerating cell lysis to release virus progeny. In contrast, direct p53 target p21 inhibits the cell cycle and, thereby, viral DNA replication. Silencing cyclin-dependent kinase inhibitors p21 or p27 through RNA interference (RNAi) alleviates cell cycle arrest and, thus, promotes viral DNA replication. Potential other means of promoting CRAd propagation include silencing of p53 inhibitors, which are known to inhibit CRAd replication, and silencing of p53 transcriptional target microRNA (miR)-34a. Molecules in red depict confirmed inhibitors of CRAd efficacy; molecules in green depict confirmed stimulators of CRAd efficacy; molecules in blue depict proposed, but not yet experimentally confirmed modulators of CRAd efficacy; orange delineated viral processes can be influenced to increase potency; cellular processes that are affected by viruses during infection and that can be modulated by transgenes and RNAi molecules expressed by armed CRAds are delineated in black.

**Figure 2 cells-07-00228-f002:**
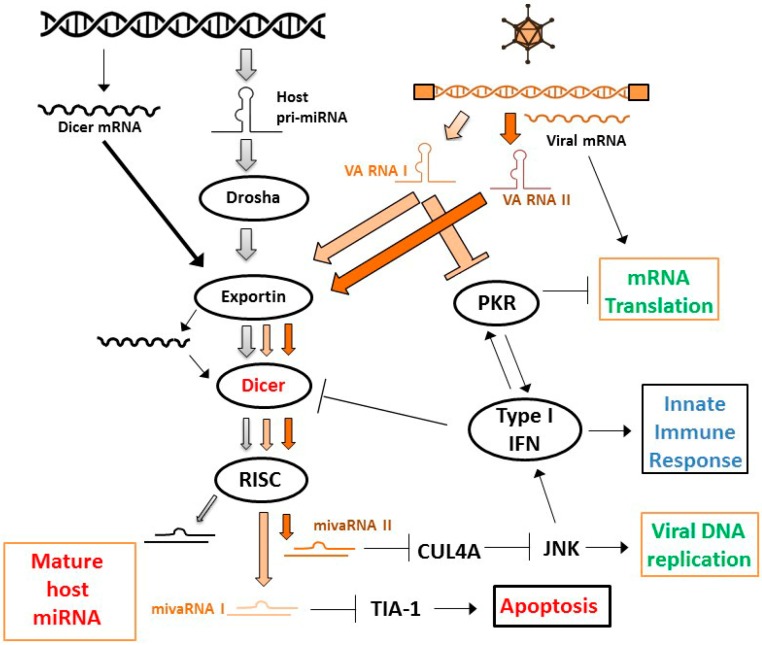
Interaction between adenovirus-encoded small RNA molecules and the host cell RNAi machinery and interferon–double-stranded RNA-activated kinase (PKR) response in the late phase of infection. Virus-associated (VA) RNAs I and II inhibit endogenous miRNA processing via competition at several levels in the RNAi machinery, as well as by reducing Dicer messenger RNA (mRNA) nuclear export. Upon processing of VA RNAs into viral miRNAs (mivaRNAs), they silence cellular mRNAs to inhibit apoptosis and promote viral DNA replication. VA RNA I directly inhibits PKR to promote (viral) mRNA translation and, thus, protein expression for capsid production. Regulation of type I interferons evoking innate immune responses is complex, with on one hand mivaRNA II-mediated stimulation and on the other hand VA RNA I-mediated inhibition of PKR-dependent stimulation. The type I interferon response inhibits Dicer function, contributing to inhibition of miRNA processing. Colors illustrate the different effects in cells in the late phase of infection: red, inhibited; green, stimulated; blue, complex regulation; orange, virus-encoded and virus-related; black, cellular.

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
