# Peer review of "Unleashing the Full Potential of Oncolytic Adenoviruses against Cancer by Applying RNA Interference: The Force Awakens"

_cells, 2018, doi:10.3390/cells7120228_

Round 1

Reviewer 1 Report

Brachtlova and van Beusechem review the application of RNA interference in the context of oncolytic adenovirus therapy. After providing an introduction to oncolytic virotherapy in general and adenoviruses specifically, they describe different strategies to improve this treatment with RNAi.

Overall, the manuscript is comprehensive and of interest to those working in the field of oncolytic virotherapy and adenovirus vector engineering.  

I have only a few comments and suggestions:

Line 53: It may be worth mentioning the type of viruses that are approved, i.e. HSV in the US and Europe and adenovirus in China, especially since this work focuses on adenoviruses.

Line 59: The poliovirus strain currently used in oncolytic therapy was attenuated by passaging and genetically modified.

The structure of the paragraphs does not seem intuitive to me.  Currently, the structure is:

1.       Oncolytic virotherapy

2.       Oncolytic Adenoviruses

2.1 More effective delivery

2.2. Improving specificity with miRNAs

2.3 Improving potency

3.       Gene suppression to make oncolytic viruses more effective

3.1   Suppression of CRAd-inhibitory genes

3.2   Exploiting virus-host interactions via microRNAs

3.3   RNAi screening

Perhaps a different structure is more logical, for instance:

1.       Oncolytic virotherapy

1.1 Oncolytic Adenoviruses

2.        Tumor targeting

2.1 More effective delivery

2.2. Improving specificity with miRNAs

3.       Improving potency

3.1   Suppression of CRAd-inhibitory genes

3.2   Exploiting virus-host interactions via microRNAs

3.3   RNAi screening

Line 456: Should the „RNAi screening“ paragraph begin here already?

Line 357: The topic of immune stimulation does not seem to belong in this paragraph („Combining OVT with Suppression of CRAd-Inhibitory Target Genes in Cancer Cells“)

Line 146: What was the rationale for choosing miR-199?

Are endogenous miRNA levels generally sufficient to improve adenovirus specificity by microRNA „de-targeting“? Which miRNA expression levels or how many miRNA copies per cell are required?

Lines 168ff.: Timing and expression levels over time appear to be crucial in microRNA approaches. This could be emphasized more and the discussion on this topic expanded.

Lines 196ff.: Since it is not intuitive why p53 expression should enhance adenovirus replication, perhaps provide a more detailed explanation.

Line 281: It is suggested to use an shRNA targeting PKR, a major antiviral effector molecule. Please comment on potential implications for vector safety. This also applies for strategies targeting cell cycle inhibitors (lines 310ff.) and anti-viral microRNAs (lines 464ff.).

In addition to ex vivo cultures and novel mouse models, Syrian hamster models have recently been described to analyze immunological effects of adenovirus therapy (e.g. Wang et al., Nature Communications 2017).

Line 445: What were the effects of IL-24/ what was the rationale for using this molecule/ what is its mechanism of action?

Sections 3.2 and 3.3: microRNAs have pleiotropic effects and often multiple targets within one cell. Perhaps comment on the resulting complexity.

Thinking towards clinical application, are there specific implications of RNAi in oncolytic virotherapy in terms of pharm/tox studies and regulatory approval?

Minor language corrections are necessary, e.g. lines 185/192 „suicide approaches“ and „suicide“ are ambiguous; line 240 „intent“ not „intend“, line 384 no space before comma, line 400 „viral gene expression“ not „virus gene expression“; line 590 „that is perfected every day“ does not seem appropriate.

Author Response

Manuscript ID cells-390469

Reviewer #1:

Brachtlova and van Beusechem review the application of RNA interference in the context of oncolytic adenovirus therapy. After providing an introduction to oncolytic virotherapy in general and adenoviruses specifically, they describe different strategies to improve this treatment with RNAi.

Overall, the manuscript is comprehensive and of interest to those working in the field of oncolytic virotherapy and adenovirus vector engineering. 

I have only a few comments and suggestions:

Line 53: It may be worth mentioning the type of viruses that are approved, i.e. HSV in the US and Europe and adenovirus in China, especially since this work focuses on adenoviruses.

            Reply: Done. See new text on lines 52-54 of the revised manuscript.

Line 59: The poliovirus strain currently used in oncolytic therapy was attenuated by passaging and genetically modified.

Reply: The reviewer is correct. See new text on lines 59-66 of the revised manuscript with marked changes; lines 59-65 of the clean version.

The structure of the paragraphs does not seem intuitive to me.  Currently, the structure is: […] Perhaps a different structure is more logical, for instance: […]

Reply: We agree that a different structure is better. We have modified the structure, where original section 2 is now subsection 1.1; and where a new section 2 “Strategies to Increase the Efficacy of Oncolytic Virus Therapy with CRAds” is now placed over original subsections sections 2.1-2.3. In addition, a new subsection 3.2 is inserted (see comment below).

 Line 456: Should the „RNAi screening“ paragraph begin here already?

Reply: This comment probably refers to the paragraph starting with “Apart from” on line 460 of our original manuscript. We understand the suggestion, but we don’t think the “RNAi screening” section should start here. That section discusses screens done to identify inhibitors of oncolytic virus efficacy. The paragraph starting at line 460 describes knowledge on virus-host miRNA interactions obtained through functional screens. The identified miRNAs are not necessarily virus inhibitors. In fact, the most interesting miRNAs with potential utility to enhance the efficacy of OVT found in these screens are those that promote virus replication.

Line 357: The topic of immune stimulation does not seem to belong in this paragraph („Combining OVT with Suppression of CRAd-Inhibitory Target Genes in Cancer Cells“)

Reply: We agree. We have inserted a separate subsection “3.2. Combining OVT with Targeting Immune Suppression” in our revised manuscript (line 403/399 in the marked/clean manuscript).

Line 146: What was the rationale for choosing miR-199?

Reply: Although the rationale was already given in the preceding sentence, we have made this more clear by inserting the new text “, which is downregulated in cancer cells,” in line 152 of the revised manuscript with marked changes; line 151 of the clean version.

Are endogenous miRNA levels generally sufficient to improve adenovirus specificity by microRNA „de-targeting“? Which miRNA expression levels or how many miRNA copies per cell are required?

Reply: From the perspective of designing novel CRAds, the important question is how many miRNA target sites should be incorporated to prevent virus replication in non-malignant cells. Clearly, there is no general answer to this question. All we can say on the basis of the available literature is that a stretch of four tandem target sites was sufficient, for the miRNAs and cell lines studied. We have inserted the following new sentence in lines 162-164 of the revised manuscript with marked changes; lines 160-162 of the clean version: “For this, insertion of four miRNA target sites appeared sufficient, but this might depend on host miRNA expression levels and the strength of the viral gene promoter.”

Lines 168ff.: Timing and expression levels over time appear to be crucial in microRNA approaches. This could be emphasized more and the discussion on this topic expanded.

Reply: Indeed this is also in our opinion crucial. We have changed the text of original lines 167-171 to: “The inhibition of virus replication that could be achieved in this study appeared inferior to that achieved when relying on host miRNAs. This can possibly be explained by the fact that the latter are already highly expressed in the host cell at the time of infection, whereas CRAd-encoded miRNAs need to be produced after cell entry and their level of expression might be replication-dependent, as copy numbers increase tremendously during replication. Hence, functional knockdown of viral genes by virus-encoded miRNAs is probably not reached before viral replication is initiated.” (lines 175-181 of the revised manuscript with marked changes; lines 173-179 of the clean version).

Lines 196ff.: Since it is not intuitive why p53 expression should enhance adenovirus replication, perhaps provide a more detailed explanation.

Reply: Following this suggestion, we have extended the text of original lines 197-200, now reading: “In contrast, expression of tumor suppressor protein p53 has not met with this limitation, presumably because adenovirus-encoded E1B55K protein interacts with p53 and regulates its activity during adenovirus replication [54,55]. In the late phase of the virus life cycle, p53 promotes cell death, thereby accelerating virus progeny release [54,56]. In addition, p53 increases late adenoviral gene expression, presumably by enhancing transcription from the virus Major Late Promoter (MLP) [57] (Figure 1).” (lines 209-215 of the revised manuscript with marked changes; lines 207-212 of the clean version). In addition, by request of reviewer #2, we added new Figure 1 that illustrates the known and suggested effects of modulating p53 activity by p53 gene expression or RNAi of p53 pathway genes on adenovirus propagation.

Line 281: It is suggested to use an shRNA targeting PKR, a major antiviral effector molecule. Please comment on potential implications for vector safety. This also applies for strategies targeting cell cycle inhibitors (lines 310ff.) and anti-viral microRNAs (lines 464ff.).

Reply: Clearly, the safety of any modification in the CRAd genome that enhances its cell killing potency depends on the selective replication properties of the virus in cancer cells. To emphasize this, we inserted a new sentence at the start of section 2.3 (lines 183-185 of the revised manuscript with marked changes; lines 181-183 of the clean version), reading: “The strong selectivity of CRAd replication in cancer cells that is obtained with currently available technology, providing excellent safety, allows exploration of methods to enhance the cancer cell killing potency of CRAds.”

In addition to ex vivo cultures and novel mouse models, Syrian hamster models have recently been described to analyze immunological effects of adenovirus therapy (e.g. Wang et al., Nature Communications 2017).

Reply: Syrian hamsters are indeed semi-permissive for human adenovirus replication and can be used to study the efficacy of CRAds in an immune competent animal model. However, there are very few tools, in particular antibodies to detect immune cell subsets, available for this animal. Nevertheless, we agree that the model should be mentioned in our review. Therefore, we inserted the following new text in lines 438-442 of the revised manuscript with marked changes; lines 434-438 of the clean version: “Although Syrian hamsters are semi-permissive for human adenovirus replication and thus provide a useful immune competent animal model for CRAd efficacy studies [98]; and CRAds armed with immune stimulating transgenes were successfully tested in these animals (e.g., [99,100]), only few antibodies are available to study hamster immune cell subsets, precluding in-depth analysis of immune responses.”

Line 445: What were the effects of IL-24/ what was the rationale for using this molecule/ what is its mechanism of action?

Reply: Although the focus here is on the miRNA inserted in the virus genome, we have added the sentence “The cytokine IL-24 was chosen for its pro-apoptotic and anti-angiogenic properties.” in lines 500-501 of the revised manuscript with marked changes; lines 495-496 of the clean version, to provide the rationale for its incorporation in the CRAd genome.

Sections 3.2 and 3.3: microRNAs have pleiotropic effects and often multiple targets within one cell. Perhaps comment on the resulting complexity.

Reply: We have followed this suggestion and inserted the following new text in lines 479-484 of the revised manuscript with marked changes; lines 474-479 of the clean version: “As miRNAs usually have many targets, modulation of a single host miRNA by a virus can have complex biological effects, silencing multiple genes that together impact virus replication. In addition, not all target genes predicted on the basis of sequence complementarity are validated as genuine miRNA targets. They may not or only under certain circumstances be silenced. This makes it very difficult to dissect virus-host interactions via miRNAs and identify the crucial host genes contributing to virus biology.”

Thinking towards clinical application, are there specific implications of RNAi in oncolytic virotherapy in terms of pharm/tox studies and regulatory approval?

Reply: We do not think so. CRAd-RNAi products will have to undergo the same biodistribution and toxicity evaluations that are required for clinical development of any CRAd. Regulatory authorities will also ask for data showing functional knockdown of the intended targets, similar to CRAds armed with transgenes, where functional protein expression needs to be shown. We do not foresee any special requirements that calls to be addressed in our review.

Minor language corrections are necessary, e.g. lines 185/192 „suicide approaches“ and „suicide“ are ambiguous; line 240 „intent“ not „intend“, line 384 no space before comma, line 400 „viral gene expression“ not „virus gene expression“; line 590 „that is perfected every day“ does not seem appropriate.

Reply: (1) Original manuscript line 186/191: The correct term is “suicide gene therapy”. At line 191 this was already correct, at line 186 (now line 198 in the marked version and 195 in the clean version) this was corrected. We also changed “suicide” into “death” in original line 194 (now line 206/203); (2-4) Original manuscript lines 242, 387, 403: corrected; (3) Original manuscript line 595: Indeed inappropriate; deleted.

Reviewer 2 Report

Adenoviruses are one of the most studied oncolytic viruses for cancer treatment. The authors have nicely summarized  and presented the knowledge of virus-host interactions that used to improve virotherapy. The authors can present some of the aspects in Tables or figures. For example, section 3.1: a table with the target genes, functions and how they were targeted will help the readers. This can be done for other sections also. This will greatly help the readers. 

Author Response

Manuscript ID cells-390469

Reviewer #2

Adenoviruses are one of the most studied oncolytic viruses for cancer treatment. The authors have nicely summarized  and presented the knowledge of virus-host interactions that used to improve virotherapy. The authors can present some of the aspects in Tables or figures. For example, section 3.1: a table with the target genes, functions and how they were targeted will help the readers. This can be done for other sections also. This will greatly help the readers.

Reply: We decided not to include Tables in the different sections. While we cover, as far as we know, all relevant reports in some sections, this is certainly not the case in others. There, we selected papers that illustrate the concepts. Presentation of the discussed target genes in tables would suggest comprehensiveness of the literature review; and is, therefore, misleading.

Following the suggestion of the reviewer, we have, however, included two Figures in our revised manuscript. Figure 1 illustrates the known and suggested effects of modulating p53 activity by expressing a p53 transgene (section 2.3) or through RNAi of p53 pathway genes (section 3.1) on adenovirus propagation. Figure 2 shows the effects of adenovirus-associated RNAs and their encoded viral miRNAs on the host RNAi machinery and IFN-PKR response (section 3 and 3.3).